# Fucoxanthin Prevents Pancreatic Tumorigenesis in C57BL/6J Mice That Received Allogenic and Orthotopic Transplants of Cancer Cells

**DOI:** 10.3390/ijms222413620

**Published:** 2021-12-19

**Authors:** Wataru Murase, Yukino Kamakura, Serina Kawakami, Ayaka Yasuda, Momoka Wagatsuma, Atsuhito Kubota, Hiroyuki Kojima, Tohru Ohta, Mami Takahashi, Michihiro Mutoh, Takuji Tanaka, Hayato Maeda, Kazuo Miyashita, Masaru Terasaki

**Affiliations:** 1School of Pharmaceutical Sciences, Health Sciences University of Hokkaido, 1757 Kanazawa, Ishikari-Tobetsu, Hokkaido 061-0293, Japan; s161155@hoku-iryo-u.ac.jp (W.M.); s161039@hoku-iryo-u.ac.jp (Y.K.); s161043@hoku-iryo-u.ac.jp (S.K.); s171164@hoku-iryo-u.ac.jp (A.Y.); s171173@hoku-iryo-u.ac.jp (M.W.); atsuhito_k@hoku-iryo-u.ac.jp (A.K.); hirokojima@hoku-iryo-u.ac.jp (H.K.); 2Advanced Research Promotion Center, Health Sciences University of Hokkaido, 1757 Kanazawa, Ishikari-Tobetsu, Hokkaido 061-0293, Japan; ohta@hoku-iryo-u.ac.jp; 3Central Animal Division, National Cancer Center, 5-1-1 Tsukiji, Chuo-ku, Tokyo 104-0045, Japan; mtakahas@ncc.go.jp; 4Department of Molecular-Targeting Prevention, Graduate School of Medical Science, Kyoto Prefectural University of Medicine, Kyoto 602-8566, Japan; mimutoh@koto.kpu-m.ac.jp; 5Department of Diagnostic Pathology and Research Center of Diagnostic Pathology, Gifu Municipal Hospital, Gifu 500-8513, Japan; tmntt08@gmail.com; 6Faculty of Agriculture and Life Science, Hirosaki University, Aomori 036-8561, Japan; hayatosp@hirosaki-u.ac.jp; 7Center for Industry-University Collaboration, Obihiro University of Agriculture and Veterinary Medicine, Hokkaido 080-8555, Japan; miyashitak@obihiro.ac.jp

**Keywords:** fucoxanthin, carotenoid, cancer chemoprevention, pancreatic cancer, CCL21, CCR7

## Abstract

Fucoxanthin (Fx) is a marine carotenoid with anti-inflammatory and anti-cancer properties in various animal models of carcinogenesis. However, there is currently no information on the effects of Fx in animal models of pancreatic cancer. We investigated the chemopreventive effects of Fx in C57BL/6J mice that received allogenic and orthotopic transplantations of cancer cells (KMPC44) derived from a pancreatic cancer murine model (*Ptf1a*^Cre/+^; *LSL*-*kras^G12D/+^*). Using microarray, immunofluorescence, western blot, and siRNA analyses, alterations in cancer-related genes and protein expression were evaluated in pancreatic tumors of Fx-administered mice. Fx administration prevented the adenocarcinoma (ADC) development of pancreatic and parietal peritoneum tissues in a pancreatic cancer murine model, but not the incidence of ADC. Gene and protein expressions showed that the suppression of chemokine (C-C motif) ligand 21 (CCL21)/chemokine receptor 7 (CCR7) axis, its downstream of Rho A, B- and T-lymphocyte attenuator (BTLA), N-cadherin, αSMA, pFAK(Tyr^397^), and pPaxillin(Tyr^31^) were significantly suppressed in the pancreatic tumors of mice treated with Fx. In addition, *Ccr7* knockdown significantly attenuated the growth of KMPC44 cells. These results suggest that Fx is a promising candidate for pancreatic cancer chemoprevention that mediates the suppression of the CCL21/CCR7 axis, BTLA, tumor microenvironment, epithelial mesenchymal transition, and adhesion.

## 1. Introduction

Pancreatic cancer (PC), a representative intractable cancer, is the seventh and fourth leading cause of cancer-related mortality worldwide and in the USA, respectively [1,2]. In the USA, it is expected that the mortality rate of PC will continue to rise for the next 20 years and become the second leading cause of cancer death by 2040 [3]. Pancreatic ductal adenocarcinoma (PDAC) accounts for approximately 90% of all cases of PC. The other PCs are comprised of those derived from neuroendocrine (5%) and acinar cells (each ≤2% in three types other than pancreatoblastoma of children) [4]. It is noteworthy that PC is frequently found as a late clinical symptom with chemotherapy resistance, aggressive growth, and metastatic dissemination when diagnosed. Therefore, many patients diagnosed are surgically unresectable. The overall 5-year survival rate remains very low (10%) [5].

Increasing evidence has demonstrated that among the overall genetic landscape, somatic mutations of four driver genes (*KRAS* oncogene, *CDKN2A*, *TP53*, and *SMAD4* tumor suppressor genes), which are all key molecules in inflammation and carcinogenesis, were remarkably characterized in many patients with PC [6,7,8,9]. Pancreatic intraepithelial neoplasia (PanIN) is recognized as a non-invasive precursor lesion for PDAC with many genetic and protein alterations. PanIN represents the stepwise progress classified into four types, from mild to severe. Genetic mutations in *KRAS*, *CDKN2A*, *TP53*, and *SMAD4* appear during the stepwise progression in PanIN and are suggested to contribute to the onset of PDAC [4,10]. In addition, mutations in other genes and aberrant regulation of mRNAs, microRNAs, non-coding RNAs, protein pathways, immune networks, tumor microenvironment (TME), epithelial mesenchymal transition (EMT), and microbiome have been identified in patients with PDAC [6,8,9,11,12,13]. To date, various genetically-engineered rodent models have been developed to elucidate the molecular mechanisms underlying carcinogenesis in PC. A PDAC murine model with the mutant *K-ras^G12D^* expression generated using the *Ptf1a* (*p48*) promoter-mediated Cre-loxP technology, *Ptf1a*^Cre/+^; *LSL*-*kras^G12D/+^* mice can trigger PanIN promotion and generate a low frequency of primary PDAC recapitulated human PDAC in long-term breeds [14]. Furthermore, the *Ptf1a*^Cre/+^; *LSL*-*kras^G12D/+^* mice combined with *Tgfbr2* knockout or *Trp53^R172H^* mutation exhibited higher frequencies of incidence and mortality in PDAC, compared with the original *Ptf1a*^Cre/+^; *LSL*-*kras^G12D/+^* mice [15,16].

Fucoxanthin (Fx) is one of the marine carotenoids contained abundantly in brown algae and diatoms. Its compound (C_42_H_58_O_6_, 658.9 g/mol) possesses a characteristic allene, a 5,6-monoepoxide, a carbonyl group, an acetyl group and two carboxyl groups (Figure 1). The Fx contents of representative edible brown algae in the world are 0.3–6.2 mg Fx/g dry weight (dw) for *Undaria pinnatifida* (Japanese name, wakame), 0.8–10.8 mg Fx/g dw for *Sargassum horneri* (Japanese name, akamoku), and 0.3–18.6 mg Fx/g dw for *Himanthalia elongata* (Sea spaghetti) [17]. Toxicological tests have revealed that Fx is a safe carotenoid with no side effects in humans and rodents [18,19,20]. Dietary Fx is metabolically converted to the deacetylated type of fucoxanthinol (FxOH) in the intestine and circulated as the main metabolite in the blood of humans and mice [21,22]. Remarkably, Fx and FxOH exerted anti-inflammatory and/or anticancer effects. Shiratori et al. showed that Fx treatment induced anti-inflammatory effects in lipopolysaccharide-treated RAW 264.7 cells by downregulating cyclooxygenase-2, inducible nitric oxide synthase, nitric oxide, prostaglandin E2, and tumor necrosis factor-α (TNF-α) [23]. In addition, many researchers have demonstrated that Fx and FxOH possess anti-inflammatory and anti-cancer effects on various cancer types in vitro and in vivo [24,25,26,27,28].

Oral ingestion of Fx could suppress obesity and diabetes, both of which are risk factors of pancreatic carcinogenesis, in human volunteers through decreases in body weight, waist circumference, and in blood triacylglycerol, C-reactive protein, and HbA1c levels [29,30]. To date, no interventional studies have been reported on PC prevention using Fx or FxOH. On the other hand, several researchers have revealed that Fx and FxOH have anti-cancer effects in PC cells. Terasaki et al. have shown that FxOH treatment induces apoptosis by attenuating chemokine, adhesion, PI3K/AKT, MAPK, and cell cycle signals in PDAC cells (KMPC44) established from PC tissue on *Ptf1a*^Cre/+^; *LSL*-*kras^G12D/+^* mice [28]. Similarly, FxOH had an apoptotic potential in PDAC cells (HaPC-5) from *N*-nitrosobis(2-oxopropyl)amine (BOP)-induced hamster PDAC model. The FxOH-treated HaPC-5 cells showed suppressed chemokine, adhesion, PI3K/AKT, and so on [27]. However, little information is available on the anti-inflammatory and anti-cancer effects of Fx in animal PDAC models.

In the present study, we investigated the suppressive effects of Fx on inflammation and cancer in a murine pancreatic cancer model using KMPC44 cells.

## 2. Results

### 2.1. Cancer Chemopreventive Effect of Fx on a Pancreatic Cancer Model Mouse with Allogenic and Orthotopic Transplantations of KMPC44 Cells

Fx (0.3%) and control diets were given to the mice ad libitum for 3 weeks (Figure 2). No clinical symptoms were observed in groups 1 or 2 during the diet administration period. The intake amount of the Fx diet-treated mice (group 1) was almost the same level as that of the control diet-treated mice (group 2): group 1, 162.9 ± 4.8 g Fx diet/kg body weight (bw); group 2, 175.9 ± 4.9 g control diet/kg bw. The average intake of Fx in group 1 was calculated as 488.8 mg Fx/kg bw. The average liver weight of group 1 was significantly higher than that of group 2: group 1, 6.0 ± 0.2 g/kg bw; group 2, 4.9 ± 0.1 g/kg bw (** *p* < 0.01). A small significant difference in body weight changes was observed between groups 1 and 2 (Figure 3A). The incidence of pancreatic tumors with macroscopically ≥1.0-mm major axis in groups 1 and 2 was 100% (8/8 mice) and 75% (6/8 mice), respectively (no significance). The estimated tumor sizes of pancreatic, parietal peritoneum, and total tumors in group 1 were significantly decreased and/or tended to be lower in the Fx diet than in group 2 (Figure 3B,C). Pathological features revealed that the number of well-differentiated tubular adenocarcinoma (ADC) (Tub 1), moderately differentiated tubular ADC (Tub 2), poorly differentiated tubular ADC (Por), and total ADCs in pancreatic tissues in the mice significantly decreased and/or tended to be lower with the Fx diet. No significant difference was observed in the onset of parietal peritoneum ADCs between groups 1 and 2 (Figure 4A–E).

### 2.2. Altering Effect of Fx on the Transcriptome in Tumor Tissue of a Pancreatic Cancer Model Mouse

Transcriptome alterations were determined in a mouse model of pancreatic cancer with allogenic and orthotopic transplantations of KMPC44 cells after 3 weeks of Fx administration. As a result, the PCoA plot indicated a clear genetic difference between groups 1 and 2 (Figure 5A). The hierarchical cluster on the 174 identified genes showed major differences in several genes between groups 1 and 2 (Figure 5B). Volcano plots revealed that the gene fold-change and *p*-values were almost the same in both up- and downregulated genes in group 1 compared to group 2 (Figure 5C). There were 86 upregulated and 88 downregulated genes (a total of 174 genes) in Fx-administered mice compared with control mice (Figure 5D). No significant bias was observed in the gene sets of mice administered Fx. Seven upregulated genes (≥2.0-fold), *Ccl24* (4.8-fold), *Ccl17* (3.9-fold), *S100a8* (2.9-fold), *Tarm1* (2.7-fold), *Anxa10* (2.4-fold), *Ptgs2* (2.0-fold), and *Ccl22* (2.0-fold) were detected as inflammatory and cancer-promoting genes. However, the upregulated genes associated with cancer chemoprevention were almost never found (Table 1, Appendix A). Seven downregulated genes (≤ −2.0-fold), *Ccl21a* (−2.4-fold), *Mmrn1* (−3.0-fold), *Dpt* (−2.7-fold), *Bank1* (−2.6-fold), *Epha7* (−2.3-fold), *Btla* (−2.2-fold), and *Vtn* (−2.1-fold) were identified as genes involved in the suppression of inflammation and carcinogenesis by Fx administration (Table 2 and Appendix A).

**Table 1 ijms-22-13620-t001:** Profile of upregulated genes in pancreatic tumors of mice treated with fucoxanthin (Fx) ^1^.

Gene Symbol	Description	Fold Change ^2^	*p*-Value ^3^
*Chil4*	Chitinase-like 4	25.77	0.0001
*Chil3*	Chitinase-like 3	18.43	0.0009
*Clca1*	Chloride channel accessory 1	5.20	0.0069
*Ccl24*	Chemokine (C-C motif) ligand 24	4.80	0.0041
*Ccl17*	Chemokine (C-C motif) ligand 17	3.87	0.0032
*Clca3b*	Chloride channel accessory 3B	3.32	0.0368
*Sprr1a*	Small proline-rich protein 1A	3.03	0.0132
*S100a8*	S100 calcium binding protein A8 (calgranulin A)	2.94	0.0017
*Duox2*	Dual oxidase 2	2.88	0.0072
*Tarm1*	T cell-interacting, activating receptor on myeloid cells 1	2.72	0.0089
*Duoxa2*	Dual oxidase maturation factor 2	2.68	0.0046
*Anxa10*	Annexin A10	2.43	0.0219
*Ppp1r35*	Protein phosphatase 1, regulatory subunit 35	2.31	0.0293
*F7*	Coagulation factor VII	2.21	0.0218
*A630095E13Rik*	RIKEN cDNA A630095E13 gene	2.17	0.0211
*Aldh1a3*	Aldehyde dehydrogenase family 1, subfamily A3	2.13	0.0205
*Creb3l1*	cAMP responsive element binding protein 3-like 1	2.06	0.0024
*Krt19*	Keratin 19	2.06	0.0337
*Ptgs2*	Prostaglandin-endoperoxide synthase 2	2.03	0.0219
*Ccl22*	Chemokine (C-C motif) ligand 22	2.02	0.0023

^1^ Among all 86 upregulated genes, 20 upregulated genes with ≥2.0-fold change were shown. ^2^ Fold change in gene expression in pancreatic tumors of mice with Fx administration (group 1) compared to that of control mice (group 2). ^3^ Significant difference between groups 1 and 2 by exact test on edge R.

**Table 2 ijms-22-13620-t002:** Profile of downregulated genes in pancreatic tumors of mice treated with fucoxanthin (Fx) ^1^.

Gene Symbol	Description	Fold Change ^2^	*p*-Value ^3^
*Prss3*	Protease, serine 3	−8.03	0.0439
*Klk1b5*	Kallikrein 1-related peptidase b5	−6.31	0.0491
*Gc*	Group specific component	−5.52	0.0169
*Klk1b4*	Kallikrein 1-related peptidase b4	−5.36	0.0476
*Ccl21a* ^4^	Chemokine (C-C motif) ligand 21A (serine)	−3.98	0.0493
*Mmrn1*	Multimerin 1	−2.99	0.0329
*Dpt*	Dermatopontin	−2.71	0.0464
*Bank1*	B cell scaffold protein with ankyrin repeats 1	−2.57	0.0092
*Xlr4b*	X-linked lymphocyte-regulated 4B	−2.54	0.0193
*Gkn3*	Gastrokine 3	−2.53	0.014
*Tbx1*	T-box 1	−2.49	0.0094
*Cd209f*	CD209f antigen	−2.43	0.0338
*Epha7*	Eph receptor A7	−2.34	0.0014
*Slc30a8*	Solute carrier family 30 (zinc transporter), member 8	−2.29	0.0349
*Cd19*	CD19 antigen	−2.27	0.0107
*Egflam*	EGF-like, fibronectin type III and laminin G domains	−2.26	0.0008
*Btla*	B and T lymphocyte attenuator	−2.15	0.014
*5330417C22Rik*	RIKEN cDNA 5330417C22 gene	−2.11	0.0003
*Kmo*	Kynurenine 3-monooxygenase (kynurenine 3-hydroxylase)	−2.09	0.025
*Klk1b8*	Kallikrein 1-related peptidase b8	−2.09	0.025
*Cpn1*	Carboxypeptidase N, polypeptide 1	−2.09	0.0246
*Vtn*	Vitronectin	−2.07	0.0409
*Efcab3*	EF-hand calcium binding domain 3	−2.06	0.0237
*Klk1b3*	Kallikrein 1-related peptidase b3	−2.06	0.011
*Triqk*	Triple QxxK/R motif containing	−2.05	0.0077
*Ntrk2*	Neurotrophic tyrosine kinase, receptor, type 2	−2.00	0.0052

^1^ Among all 88 downregulated genes significantly altered, 26 downregulated genes with ≥2.0-fold change were shown. ^2^ Fold change in gene expression in pancreatic tumors of mice with Fx administration (group 1) compared to that of control mice (group 2). ^3^ Significant differences between groups 1 and 2 by an exact test on edge R. ^4^ Genes colored gray were the genes that were examined for protein expression as significant cancer-related molecules (Figure 6A,B).

### 2.3. Altering Effect of Fx on the Protein Expressions in Tumor Tissue of a Pancreatic Cancer Model Mouse

Based on transcriptome alterations, the effect of Fx on protein expression and activation was determined in pancreatic tumors of mice. Fx administration significantly decreased CCL21 (0.6-fold), BTLA (0.3-fold), CCR7 (0.5-fold), Rho A (0.5-fold), N-cadherin (0.6-fold), αSMA (0.4-fold), pFAK(Tyr^397^) (0.4-fold), and pPaxillin(Tyr^31^) (0.3-fold). No changes in expression or activation were observed for MMRN1, DPT, BANK1, EPHA7, pERK1/2(Thr^202^/Tyr^204^), E-cadherin, cyclin D1, cyclin D2, cyclin B1, integrin α5, integrin β1, p53, pro-caspase-3, and the active form of cleaved caspase-3 (p17/p19). Vitronectin, pAKT(Ser^473^), pSTAT3(Ser^727^), vimentin, CCR1, CCR4, and NCAM2 were not detected (Figure 6A,B). The tissue expression and distribution of two molecules, CCL21 and BTLA, were examined in the pancreatic tissue of the mice by confocal microscopy. As a result, both CCL21 and BTLA were expressed and distributed strongly to malignant tissue but not the stromal tissue in group 1, and their expression almost disappeared in group 2 (Figure 7).

### 2.4. Altering Effect of Fx on Cell Growth in KMPC44 Cells with Ccr7 Knockdown

Ccr7 gene knockdown in KMPC44 cells with DsiRNA-1 and -2 treatment for 1 or 2 d significantly decreased CCR7 protein expression compared with negative control (NC) cells (Figure 8A). The cell growth in KMPC44 cells with *Ccr7* DsiRNA-1 and -2 was significantly decreased compared with NC cells at both 1 and 2 d of incubation after the re-inoculation of KMPC44 cells with Ccr7 knockdown: absorbance value (%) of NC 0.49 ± 0.01 (100%), DsiRNA-1 0.40 ± 0.01 (83.0%), DsiRNA-2 0.30 ± 0.01 (62.3%) for 1 d; NC 1.00 ± 0.01 (100%), DsiRNA-1 0.87 ± 0.01 (87.5%), DsiRNA-2 0.69 ± 0.02 (69.1%) for 2 d (Figure 8B).

## 3. Discussion

The present study demonstrated that Fx administration significantly suppressed pancreatic lesions through alterations in some inflammatory- and cancer-associated signals. This is the first report suggesting the chemopreventive effect of Fx in a murine pancreatic cancer model.

In the present study, the Fx diet (0.3% Fx) was administered ad libitum to C57BL/6J mice with allogenic and orthotopic transplantations of KMPC44 cells for 3-week periods. The mice ingested more of the Fx diet than we expected. The average intake of Fx in group 1 was calculated as 488.8 mg Fx/kg bw; this value was 9.8~16.3-fold higher in comparison with our prior findings on colon cancer chemoprevention in mice with Fx administration (30~50 mg Fx/kg bw) [25,31,32]. However, no clinical symptoms were observed in group 1, except for an increase in liver weight (1.2-fold higher than in group 2). Beppu et al. reported that a dose of 500 or 1000 mg Fx/kg bw significantly increased liver weight in both male and female ICR mice without histological abnormality of liver tissue and without increases in plasma aminotransferase, alanine transaminase, or γ-glutamyl transpeptidase activity [20]. Therefore, although we did not examine the histological and hematological features in the liver of group 1, we speculated that the mice would rarely receive serious side effects regarding the enhancement of liver weight by Fx administration. No significant difference was found in the incidence of pancreatic tumors (≥1.0-mm major axis) between groups 1 and 2; however, the estimated tumor sizes of pancreatic, parietal peritoneum, and total tumors in group 1 were significantly decreased and/or tended to be lower than those in group 2 (Figure 3C). In addition, the number of Tub1, Tub2, Por, and total ADCs in the pancreas were significantly decreased and/or tended to be lower with the Fx diet. No significant differences in Tub1, Tub2, Por, and total ADCs in the parietal peritoneum were observed between groups 1 and 2 (Figure 4A–E). These results suggest that Fx administration caused a retardation effect of ADC in both pancreatic and parietal peritoneum ADCs in mice, rather than suppressing their onset.

Transcriptome profile and protein analysis based on the gene expressions have demonstrated that CCL21/CCR7 axis and BTLA were significantly attenuated in pancreatic malignant tissues of group 1, compared with those of group 2 (Figure 6, Figure 7 and Figure 8, and Table 2). On the other hand, our previous study showed that CCR1, CCR4, NCAM2, pFAK(Tyr^397^), pPaxillin(Tyr^31^), cyclin D1, cyclin B1, pAKT(Ser^473^), pAKT(Thr^308^), pMEK1/2(Ser^217/221^), and pERK1/2(Thr^202/Tyr204^) in FxOH-treated KMPC44 cells were downregulated, caspase-3 was activated, and apoptosis was induced (28). In addition, TME, EMT, growth, inflammation, cell cycle, adhesion (integrin signaling), and apoptosis are key regulators of human pancreatic carcinogenesis [6,8,9,11,12]. We then confirmed the expression and activation of proteins involved in apoptosis induction in KMPC44 cells, TME, and EMT. Consequently, N-cadherin, αSMA, pFAK(Tyr^397^), and pPaxillin(Tyr^31^) were downregulated in the pancreatic tumors of mice administered Fx (Figure 6A,B). Our results suggest that Fx suppresses the CCL21/CCR7 axis downstream of Rho signaling, BTLA, TME, EMT, and adhesion in pancreatic tumors in mice. The results on adhesion molecules led us to speculate that adhesion may be due to integrins other than integrin α5 and β1. The cancer chemopreventive effect and underlying regulations are summarized in Figure 9.

CCL21, a small chemoattractant cytokine, is one of two ligands (CCL21and CCL19) for the chemokine receptor CCR7. Early studies have shown that CCL21 is constitutively expressed in the lymph nodes and stimulates the migration of lymphocytes to lymphoid organs [33,34,35]. CCL21/CCR7 generally promotes the cell cycle, MAPK, JAK/STAT, NFκB, PI3K/AKT, and Rho A signals in many cancer cells with cell growth and migration [36,37,38,39,40]. In contrast, the CCL21/CCR7 axis signal also plays an important role in pancreatic cancer. CCR7 enhances the migration, invasion, EMT, tumorigenesis, and lymphatic spread in various types of pancreatic cancer cells through the regulation of ERK, PI3K/AKT, and NFκB [39,41,42,43]. CCR7 expression is highly expressed in patients with pancreatic cancer and is suggested to be positively associated with poor outcomes such as short survival, lymph node metastasis, lymphangiogenesis, and angiogenesis [41,44,45]. BTLA (CD272) has been reported as one of the CD28 immunoglobulin superfamily expressed on all lymphocytes and a type I transmembrane co-signaling receptor attenuating T cell function [46]. The ligand of the molecule is a herpesvirus entry mediator (HVEM). BTLA activation suppresses cancer-specific CD8^+^ T-cell proliferation and differentiation, and its high expression in whole cancer tissue or infiltrating T cells is associated with poor outcomes in patients with epithelial ovarian, gallbladder, and gastric cancers [47,48,49,50,51,52]. A high level of plasma BTLA correlates with poor overall survival in PDAC patients [53]. In addition, inhibition of BTLA attenuates tumorigenesis in mice [50]. Conversely, low expression of BTLA was associated with poor overall survival in patients with colorectal cancer [54]. The correlation between clinical characteristics and BTLA expression remains inconsistent depending on the cancer type. However, the functional inhibition of CCR7 and BTLA may be a key trigger for pancreatic cancer prevention. Moreover, in the present study, the expression of the tumor suppressor p53 and the activation of caspase-3 were not observed in the pancreatic tissue of Fx-treated mice (Figure 6). These results suggest that the induction of caspase-dependent apoptosis may scarcely occur in the pancreatic tumors of mice. Further investigation is needed to elucidate the molecular mechanisms underlying cancer chemopreventive effects in mice treated with Fx.

Administration of a non-polar carotenoid, β-carotene, significantly suppressed carcinogenesis in BOP-treated hamsters and azaserine-treated rats [55,56]. Other natural compounds such as fermented brown rice, 4-methylthio-3-butenyl isothiocyanate, benzyl isothiocyanate, sulforaphane, and green tea polyphenols, are also potential chemopreventive agents in the BOP-induced hamster pancreatic cancer model [55,57,58,59]. The anticancer mechanisms of natural compounds in these pancreatic cancer models remain unknown. There is little information on the CCL21/CCR7 axis, BTLA, TME, EMT, and adhesion signals on the effect of natural compounds on pancreatic cancer. Therefore, our study may be the first to suggest a function of several signals in pancreatic cancer chemoprevention using natural compounds.

## 4. Conclusions

Fx administration suppressed pancreatic tumorigenesis in a pancreatic cancer murine model. Fx significantly altered the expression of 174 genes in pancreatic tumors. Analyses of gene and protein expression based on inflammation and carcinogenesis suggested that the decreases in CCL21 and BTLA are important in cancer chemoprevention by Fx. In addition, downregulation of CCR7, Rho A, N-cadherin, αSMA, pFAK(Tyr^397^), and pPaxillin(Tyr^31^) were observed in the Fx-treated mice. Furthermore, *Ccr7* knockdown in KMPC44 cells significantly attenuated cell growth. Our findings suggest that the CCL21/CCR7 axis, BTLA, TME, EMT, and adhesion may be key regulators of cancer chemoprevention in C57BL/6J mice receiving allogenic and orthotopic transplants of KMPC44 cells by Fx administration.

## 5. Materials and Methods

### 5.1. Chemical

A highly concentrated Fx-oil (5.0% *w*/*v*) composed mainly of palm oil and some minor ingredients (proteins, carbohydrates, and sodium) was kindly donated by Oryza Oil & Fat Chemical (Aichi, Japan). The control oil without Fx was prepared by the same company. RPMI-1640 cell culture medium, Hank’s balanced salt solution (HBSS), isoflurane, and 37% formaldehyde were purchased from FUJIFILM Wako Pure Chemicals (Osaka, Japan). DynaMarker RNA High for Easy Electrophoresis was obtained from the BioDynamics Laboratory (Tokyo, Japan). RNAlater, Lipofectamine RNAiMAX, Opti-MEM I medium, GlutaMAX^TM^, and penicillin/streptomycin were purchased from Thermo Fisher Scientific (Carlsbad, CA, USA). WST-1 reagent and bovine serum albumin (BSA) were obtained from Roche Diagnostics (Manheim, Germany) and Nacalai Tesque (Kyoto, Japan), respectively. Anti-chemokine C-C ligand 21 (CCL21)/6Ckine, anti-chemokine receptor 7 (CCR7)/CD197, anti-cyclin D2, and anti-NCAM2 antibodies were purchased from Bioss Antibodies (Beijing, China). Anti-multimerin 1 (MMRN1), anti-B cell scaffold protein with ankyrin repeats (BANK1), anti-dermatopontin (DPT), and anti-RhoA antibodies were obtained from Novus Biologicals (Littleton, CO, USA). Anti-CCR1 and anti-CCR4 antibodies were purchased from BioVision (Milpitas, CA, USA). Anti-ephrin A7 (EphA7) and B- and T-lymphocyte attenuator (BTLA/CD272) antibodies were purchased from Proteintech (Chicago, IL, USA) and Abeomics (San Diego, CA, USA), respectively. Anti-phosphorylated (p) Akt(Ser^473^) antibody was obtained from GenScript (Piscataway, NJ, USA). Anti-vitronectin, anti-vimentin, anti-N-cadherin, anti-E-cadherin, anti-cyclin D1, anti-cyclin B1, anti-integrin α5, anti-integrin β1, anti-pFAK(Tyr^397^), anti-caspase-3, and anti-β-actin antibodies were purchased from GeneTex (Irvine, CA, USA). Anti-pERK1/2(Thr^202^/Tyr^204^), anti-pSTAT3(Ser^727^), and anti-α-smooth muscle actin (α-SMA) antibodies were obtained from Cell Signaling Technology (Danvers, MA, USA). Anti-pPaxillin(Tyr^31^) and anti-p53 antibodies were purchased from Novex (San Diego, CA, USA). Goat anti-rabbit IgG conjugated with Alexa Fluor 488, and ProLong Gold Antifade conjugated with 4′,6-diamidino-2-phenylindole (DAPI) were obtained from Invitrogen (Carlsbad, CA, USA). All other chemicals and solvents were of high-grade quality.

### 5.2. Cell Culture

A mouse pancreatic cancer cell line, KMPC44, from pancreatic adenocarcinoma in genetically modified C57BL/6J mice with *Ptf1a*^Cre/+^; *LSL*-*k-ras^G12D/+^* was established by Dr. Mami Takahashi (National Cancer Center Research Institute, Japan) as previously described [26]. Briefly, *Ptf1a*^Cre/+^; *LSL*-*k-ras^G12D/+^* mice were produced by crossing transgenic *Ptf1a*-Cre mice with a STOCK *Ptf1a*
^tm1(cre)Cvw^ (Mutant Mouse Regional Resource Centers, Bar Harbor, ME) and transgenic LSL-*K-Ras^G12D^* mice with a 129-*Kras2^tm4Tyj^* (National Cancer Institute, Mouse Models of Human Cancers Consortium, Rockville, MD, USA). A pancreatic adenocarcinoma well differentiated in female *Ptf1a*^Cre/+^; *LSL*-*K-ras*^G12D/+^ mice (75-week-old) was cultured in RPMI-1640 medium with 10% heat-inactivated fetal bovine serum (FBS), 1% GlutaMAX^TM^, and 1% penicillin/streptomycin (10%FBS/RPMI). The pancreatic cancer cell line (KMPC44) was purified by repeating passage with the culture medium and subcloned by the limit dilution method. The experimental design was approved by the Institutional Ethics Review Committee for Animal Experimentation, followed by the Guidelines for Animal Experiments of the National Cancer Center (identification code, T13-013; authorization, 13 March 2013). The cells were routinely passaged in 10% FBS/RPMI until use.

### 5.3. Construction of Pancreatic Cancer Model Mice with Allogenic and Orthotopic Transplantations of KMPC44 Cells, and Animal Experiments

Animal experiments are illustrated in Figure 2. Five-week-old female C57BL/6J mice were purchased from Sankyo Labo Service (Tokyo, Japan), randomly divided into plastic cages (4 mice/cage) with sterilized softwood chips, and acclimated under controlled temperature, humidity, and a 12 h light/dark cycle. Standard solid MF chow (Oriental Yeast, Tokyo, Japan) and tap water were provided to the mice ad libitum. Fx oil was applied to a solid MF at a concentration of 0.3% Fx (3.0 mg Fx/g MF chow, Fx diet). The control diet was prepared as a solid MF with equivalent palm oil applied (control diet). After a week of acclimation, the mice were divided into the Fx diet-fed group (group 1, *n* = 8) and control diet-fed group (group 2, *n* = 8). The mice in groups 1 and 2 were administered Fx and control diets ad libitum for 3 weeks until sacrifice. KMPC44 cells were trypsinized, washed with phosphate-buffered saline (PBS), and suspended in cold-HBSS (concentration, 1 × 10^6^ cells/5 μL HBSS). After a week of Fx or control diet administration, the mice (7-weeks-old) were depilated, and the pancreatic tissue was pulled out by opening the abdomen under isoflurane anesthesia. Then, the suspension of KMPC44 cells (1 × 10^6^ cells) was orthotopically transplanted into the pancreatic tissue of the mice using a 27G 1/2 needle-equipped 25 μL glass microsyringe (Type 702LT, Hamilton, Las Vegas, NY, USA). After rapid suture, the mice were given Fx or a control diet for 2 weeks. The mice were sacrificed under isoflurane anesthesia, and the whole pancreatic tissue and tumor on the parietal peritoneum from each mouse was excised, washed with cold PBS, and pinned onto a rubber plate. For histopathological analyses, the pancreatic tissue and tumors in the parietal peritoneum of the mice were fixed in 10% formalin for 2 days. The tumor sizes of pancreatic and parietal peritoneal tumors in each mouse were macroscopically evaluated using formalin fixation. The estimated tumor size (mm^3^) was determined based on the following formula: a (mm) × b^2^ (mm)/2 (a, long range; b, short range). Hematoxylin-eosin and paraffin-embedded sections that fixed the pancreatic and parietal peritoneum tissue sections were prepared by Morphotechnology (Sapporo, Japan). Subsequently, other pancreatic tumors in the mice (group 1, *n* = 5; group 2, *n* = 4) were excised, washed with cold PBS, and immersed in RNAlater (500 μL) overnight at 4 °C, then frozen at −80 °C until transcriptome analysis or western blot analysis. Histopathologic examination of pancreatic lesions in mice was performed by an expert pathologist. The animal experiments adhered to the Institutional Guidelines for Animal Care and Use in the Health Sciences University of Hokkaido, as well as “Guidelines for Animal Experiments in the Health Sciences University of Hokkaido” (identification code, 20-014; authorization, 9 March 2020).

### 5.4. Microarray Analysis

Total RNA from the whole pancreatic tumor from each mouse was extracted and purified using an RNeasy Mini Kit with QIA shredder and RNase-Free DNase Set (QIAGEN, Valencia, CA, USA) in accordance with the manufacturer’s instructions. Total RNA concentration and quantification were determined using a Nanodrop^®^ ND-1000 (NanoDrop, Wilmington, DE, USA) and agarose gel electrophoresis was used to detect the diminution of ribosomal RNA 18S and 28S with DynaMarker RNA High for Easy Electrophoresis (BioDynamics Laboratory, Tokyo, Japan). Transcriptome analysis of pancreatic tumors was conducted using the “GeneChip^TM^ WT PLUS reagent kit manual target preparation for GeneChip^TM^ whole transcript (WT) expression arrays” (Thermo Fisher Scientific, Waltham, MA, USA). Total RNA (500 ng) was mixed with diluted Poly-A RNA controls, followed by synthesis of the first- and second-strand complementary DNA (cDNA). Single-strand cRNA was produced from double-stranded cDNA using an in vitro transcription method based on T7 RNA polymerase. Then, the second-cycle single-strand cDNA was transcribed from the cRNA, fragmented by uracil-DNA glycosylase and apurinic/apyrimidinic endonuclease 1, and biotin-labeled with terminal deoxynucleotidyl transferase. The labeled cDNA template was hybridized to a Clariom^TM^ S mouse array (Thermo Fisher Scientific, Carlsbad, CA, USA). The array was individually washed and stained on an Affymetrix Fluidics Station 450, scanned using the Affymetrix GeneChip Scanner 3000 system (Affymetrix, Santa Clara, CA, USA), and then analyzed using Transcriptome Analysis Console (TAC) software version 4.0.2 (Applied Biosystems, Foster City, CA, USA). The genes with significant differences between the two groups were evaluated using ≥1.5 and ≤−1.5 -fold with cutoff *p*-value [one-way analysis of variance (ANOVA), *p* < 0.05]. Bioinformatics analyses such as principal coordinate analysis (PCoA), volcano plots, and hierarchical clustering on gene expression were conducted using TAC software.

### 5.5. Western Blot Analysis

Pancreatic tumors in mice were collected and sonicated with lysis buffer, and the protein content was determined using the Bradford assay. An aliquot of the protein solution (10 μg) in each mouse was subjected to sodium dodecyl sulfate-10% polyacrylamide gel electrophoresis (SDS-PAGE), and then electroblotted onto Hybond PVDF membranes (Amersham Bioscience, Little Chalfont, UK). The PVDF membrane was incubated with Tris-buffered saline containing 0.1% Tween 20 and 1 *w*/*v*% BSA (1% BSA/TBS-T) at room temperature for 1 h and treated with each primary antibody (diluted 1:1000) in 1% BSA/TBS-T at 4 °C overnight. Subsequently, the membranes were incubated with HRP-conjugated anti-mouse or anti-rabbit secondary antibody-HRP conjugates in 1% BSA/TBS-T at room temperature for 1 h. Protein bands were detected using a chemiluminescence assay (Millipore, Billerica, MA, USA).

### 5.6. Fluorescence Immunohistochemistry

Paraffin wax-embedded sections of pancreatic tissue from each mouse were routinely washed with xylene, anhydrous ethanol, 95% ethanol aq, and distilled water in sequence. For antigen retrieval, the sections were treated with 1 mM EDTA buffer (pH 9.0) at 95 °C for 20 min and washed with PBS and TBS-T. The sections were then incubated with 5% *w*/*v* BSA/TBS-T (5% BSA/TBS-T) at room temperature for 1 h. The sections were probed with CCL21/6Ckine or BTLA rabbit polyclonal primary antibody (diluted 1:50) in 5% BSA/TBS-T at 4 °C overnight. Each section was washed with TBS-T and probed with goat anti-rabbit IgG-Alexa Fluor 488 conjugate (diluted 1:100) in TBS-T for 1 h at room temperature in the dark. Finally, the sections were washed with TBS-T and PBS and mounted with ProLong Gold antifade conjugated with DAPI. The localization of CCL21/6Ckine or BTLA in the pancreatic tissue was observed using a Nikon Eclipse Ti2 confocal microscope with a Nikon A1 laser acquisition system (Nikon, Tokyo, Japan).

### 5.7. Gene Knockdown

Twenty-seven mer of dicer-substrate short interfering RNA (dsiRNA) on the coding sequences of *Ccr7* mRNA in *Mus musculus* was designed by Integrated DNA Technologies (Coralville, IA, USA). The designed *Ccr7* dsiRNAs were as follows: dsRNA-1, 5′-CUG AUA CCU UUC CUC AUG UUC UGT T-3′, and duplex-2, 5′-GAU ACU GAC GUA CAU CUA UUU CAA G-3′, and their negative control (NC), 5′-GUG UUC UAC ACC AUU ACU CAA UUC UUA-3′. Lipofectamine RNAiMAX and Opti-MEM I medium were used to prepare the dsiRNA complexes in accordance with the manufacturer’s instructions. The KMPC44 cells were seeded in 100-mm dishes at a density of 6 × 10^4^ cells/mL and adhered for 1 d. 5.5 μL of dsiRNA or NC was mixed with 30 μL of Lipofectamine RNAiMAX^TM^ and 1000 μL of Opti-MEM I medium. The dsiRNA -1, -2, or NC complex was added to 10 mL of culture medium (final concentration of dsiRNA, 10 nM) for 1 d. Then, the medium was changed to 10% FBS/RPMI containing the new dsiRNA complex and boosted for 1 d. The cells with *Ccr7* knockdown were reinoculated in a 24-well plate at a density of 5 × 10^4^ cells/mL and adhered for 1 or 2 days. The growth of treated cells was determined using a cell viability assay.

### 5.8. Cell Viability Assay

WST-1 reagent (50 μL/well in a 24-well plate) was added to KMPC 44 cells with a gene knockdown and incubated for 4 h at 37 °C. Cell viability was measured at 450 nm using an ELISA microplate reader (Tecan Japan, Tokyo, Japan).

### 5.9. Statistics Analysis

All results are expressed as mean ± standard error (SE) for each experiment. Statistical differences were determined using Fisher’s exact probability test for tumor incidence, one-way ANOVA for whole gene expression analysis in mice between two groups, and Wilcoxon rank sum test or Student’s t-test for comparisons between two groups. Differences were considered statistically significant at * *p* < 0.05, ** *p* < 0.01, and exact *p*-values by an exact test on edge R in the TAC software.

## Figures and Tables

**Figure 1 ijms-22-13620-f001:**
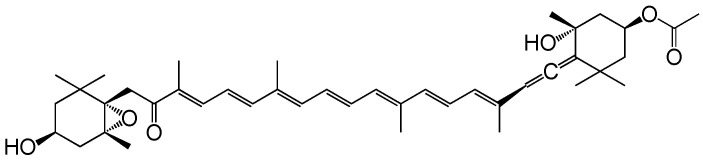
Molecular structure of fucoxanthin (Fx). Molecular weight, 658.91.

**Figure 2 ijms-22-13620-f002:**
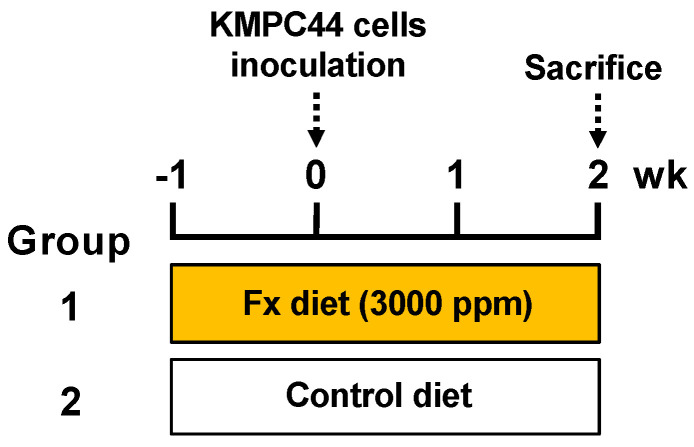
Experimental protocol for pancreatic cancer model mice with allogenic and orthotopic transplantations of KMPC44 cells. Fx diet (0.3% Fx) was given to group 1 ad libitum for 3 weeks before sacrifice (orange paint). The control mice (group 2) were given a control diet (Fx-free) (white paint).

**Figure 3 ijms-22-13620-f003:**
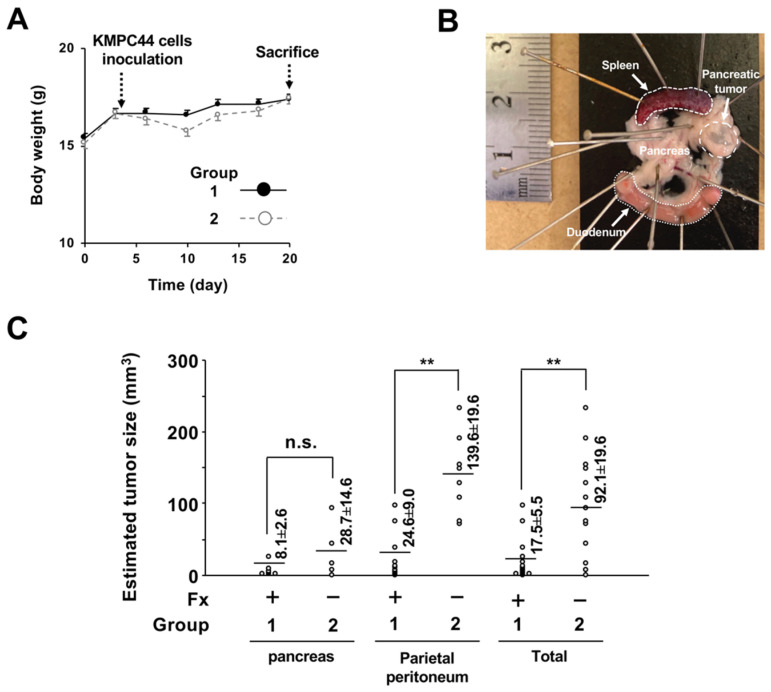
Body weight and estimated tumor size in pancreatic cancer model mice. (**A**) Body weight changes during the period of Fx diet administration. Black and white dots indicate groups 1 and 2, respectively. (**B**) Macroscopic observation of a pancreatic tumor in mice with allogenic and orthotopic transplantations of KMPC44 cells just after sacrifice (white arrow). (**C**) Estimated pancreatic tumor size in pancreatic and parietal peritoneum tissues in mice. The tumor size of tumors with ≥1 mm of major axis was estimated based on a formula: a (mm) × b^2^ (mm)/2 (a, long range; b, short range). Means ± SE (*n* = 6–21). ** *p* < 0.01. n.s., no significance.

**Figure 4 ijms-22-13620-f004:**
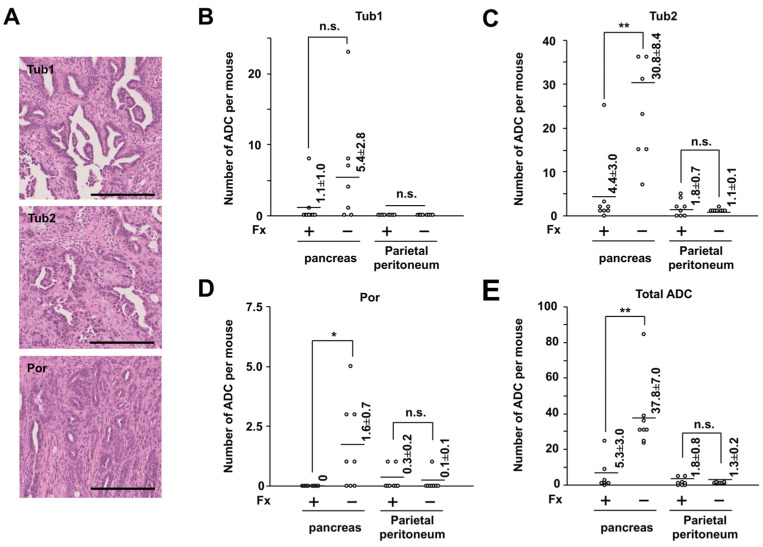
Effect of pancreatic cancer chemoprevention in pancreatic cancer model mice by fucoxanthin (Fx). (**A**) Representative histopathology of pancreatic adenocarcinoma (ADC). Bar, 200 μm. Tub1, well differentiated tubular ADC. Tub2, moderately differentiated tubular ADC. Por, poorly differentiated tubular ADC. The number of (**B**) Tub 1, (**C**) Tub 2, (**D**) Por, and (**E**) total ADCs in pancreatic and parietal peritoneum tissues. Flat bars, Mean ± SE (*n* = 8). * *p* < 0.05 and ** *p* < 0.01. n.s., no significance.

**Figure 5 ijms-22-13620-f005:**
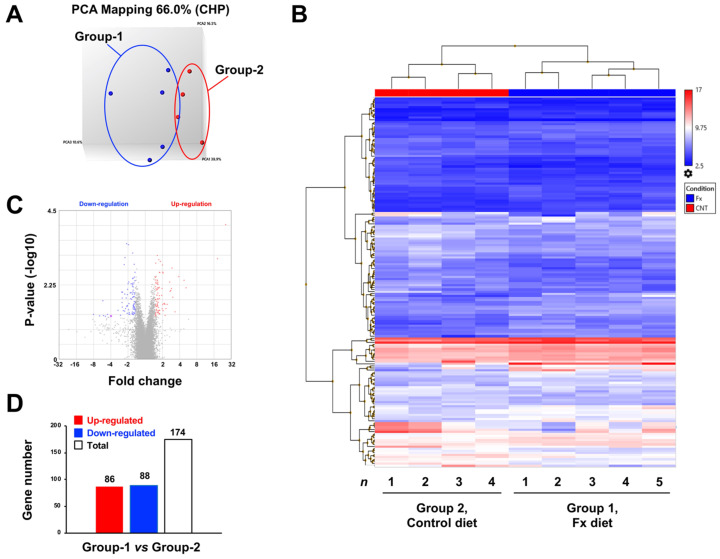
Gene expression profiles in pancreatic cancer model mice with or without fucoxanthin (Fx) administration. Gene expression levels between the Fx-treated (group 1) and control (group 2) mice were subjected to Clariom^TM^ S mouse assays and TAC software (*n* = 4–5). (**A**) PCoA plots on gene set distance between groups 1 and 2. (**B**) Hierarchical clustering analysis for 174 genes between groups 1 and 2. (**C**) Volcano plots between groups 1 and 2. (**D**) Total number of up- (≥1.5-fold) and downregulated (≤−1.5-fold) genes between groups 1 and 2.

**Figure 6 ijms-22-13620-f006:**
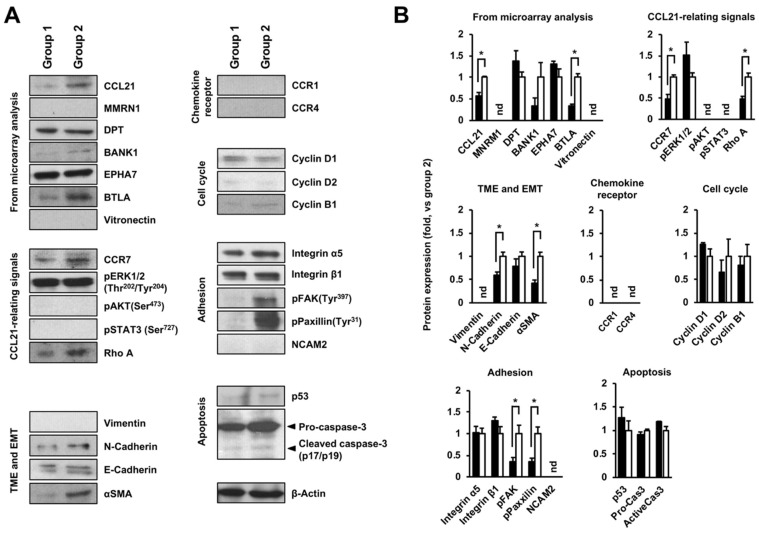
Profiles of protein expression and activation in pancreatic tumors of pancreatic cancer model mice with or without fucoxanthin (Fx) administration. Protein expression and activation levels between Fx-treated (group 1) and control (group 2) mice were evaluated using western blot analysis. (**A**) Each representative protein band in the pancreatic tumors of groups 1 and 2. (**B**) The values of each protein band of pancreatic tumor in groups 1 and 2 were normalized to that of the β-actin protein band density from the image. Each protein level in group 1 was evaluated against 1.0-fold in group 2. Mean ± SE (*n* = 4). * *p* < 0.05. nd, not detected.

**Figure 7 ijms-22-13620-f007:**
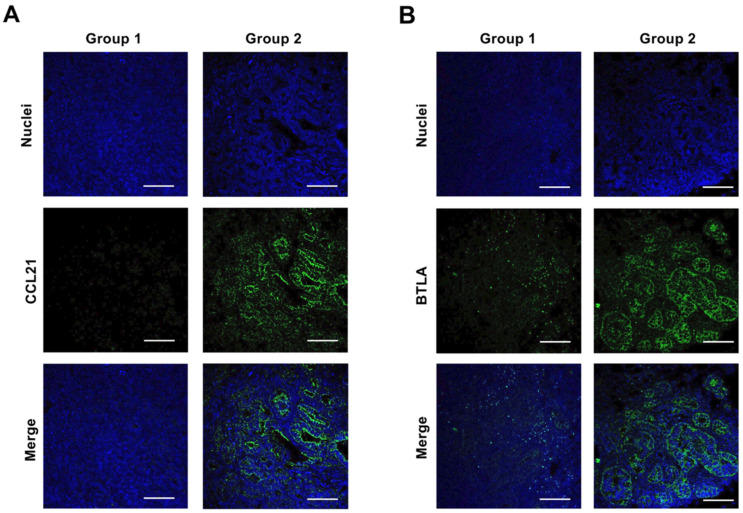
CCL21 and BTLA protein expression in tumor tissues in pancreatic cancer model mice with or without fucoxanthin (Fx) administration. Green fluorescence indicates CCL21 and BTLA. Blue fluorescence indicates nuclei. Confocal images of (**A**) CCL21 and (**B**) BTLA on tumor tissues in pancreatic cancer model mice. White bars are 100 μm.

**Figure 8 ijms-22-13620-f008:**
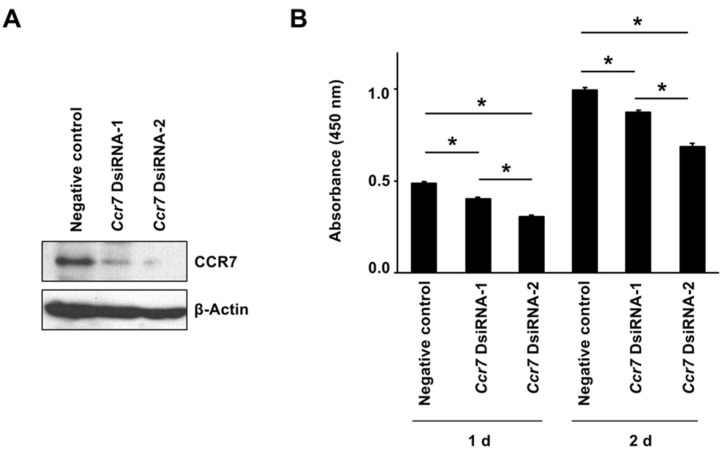
Effect of *Ccr7* knockdown on cell growth in KMPC44 cells. KMPC44 cells were inoculated in a 100-mm dish at a density of 6 × 10^4^ cells/mL, treated with *Ccr7* dicer-substrate short interfering RNA (dsiRNA) for 1 d, and then the new dsiRNA was boosted for 1 d. Subsequently, the cells with *Ccr7* knockdown were reinoculated in a 24-well plate at a density of 5 × 10^4^ cells/mL, adhered for 1 or 2 d. (**A**) CCR7 and β-actin protein expressions in KMPC44 cells with or without *Ccr7* knockdown for total 2 d were observed using western blot analysis. (**B**) The cell growth of KMPC44 cells was determined using a WST-1 cell viability assay. Mean ± SE (*n* = 6). * *p* < 0.05.

**Figure 9 ijms-22-13620-f009:**
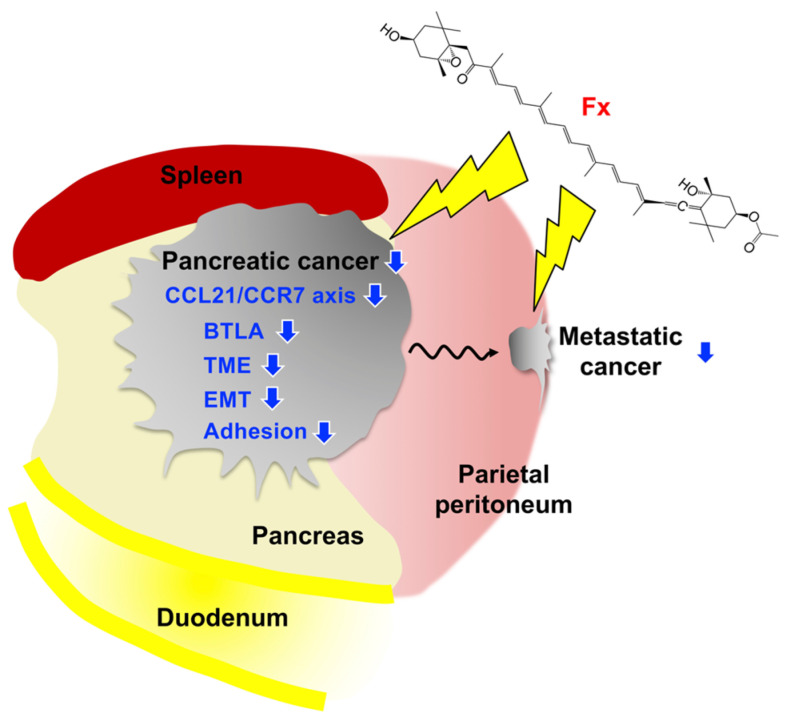
Possible mechanisms underlying the pancreatic cancer chemopreventive effects on C57BL/6J mice received allogenic and orthotopic transplantations of KMPC44 cells. Fx administration prevents tumorigenesis on pancreatic and parietal peritoneum tissues through the suppressions of the CCL21/CCR7 axis downstream of Rho A signal, BTLA, TME, EMT, and adhesion. Blue arrows show inhibition or decrease.

## Data Availability

The data in the present study are available from the corresponding author upon reasonable request.

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
