# Peer review of "Fucoxanthin Prevents Pancreatic Tumorigenesis in C57BL/6J Mice That Received Allogenic and Orthotopic Transplants of Cancer Cells"

_ijms, 2021, doi:10.3390/ijms222413620_

Round 1
Reviewer 1 Report
It is opinion of this reviewer that this valuable and well prepared article only few changes/corrections:
Title – capital first letters should be used.
84 – The reference should be cited.
91 – How Fx can suppress obesity and diabetes?
102-104 – This is like as summary. Please rephrase the aim of this research.
108 – “ppm” is not an unit of SI.
Fig. 1A and 1B should be presented as separated single Figures.
Information about the used statistical test should be removed from figures description. “4.9 Statistical Analysis” informs about used test.
Figure 4 B – Please check significance of the difference for microarray analysis for BANK 1.
235 – It should be “… activity [20].”
247-261 – This part should belong rather to “Results” instead to “Discussion”.
318 – I suggest to add “Conclusions” in this place.
539 – “Cancer Cell” with italic, “2005” with bold.
565 – Journal title abbreviation is needed.
587, 607 –It should be “PLoS ONE”.
Author Response
Dec 3, 2021
Dear Ms. Mischa Wang
Assistant Editor, International Journal of Molecular Sciences
Manuscript ID.: ijms-1483444
Ms.Title: Fucoxanthin prevents pancreatic tumorigenesis in C57BL/6J mice that received allogenic and orthotopic transplantations of the cancer cells
Thank you for your email of Nov 30, 2021, regarding our manuscript, “Fucoxanthin prevents pancreatic tumorigenesis in C57BL/6J mice that received allogenic and orthotopic transplantations of the cancer cells”, and the valuable comments of the reviewers. The changed parts in the manuscript file are marked in the red with Track Changes.
Please find enclosed the revised manuscript in Word format (file name: ijms-1483444 with Track Changes).
Comments from the editors and reviewers:
Reviewer 1: It is opinion of this reviewer that this valuable and well prepared article only few changes/corrections.
Title – capital first letters should be used.
Thank you for your constructive comments. In accordance with the Reviewer-1’s comments, we have modified the title (Page 1, line 2 – 4).
84 – The reference should be cited.
Thank you for your valuable comments. In accordance with the Reviewer-1’s comments, we have added two references [21,22] (Page 2, line 86; Page 16, Ref.21 and 22). With this modification, we have changed the references number from 21-57 to 23-59.
91 – How Fx can suppress obesity and diabetes?
We express our deep thanks for your comments. In accordance with the Reviewer-1’s comments, we have modified the sentence (Page 2, line 94- Page 3, line 96).
102-104 – This is like as summary. Please rephrase the aim of this research.
Thank you for finding the mistake. In accordance with the Reviewer-1’s comments, we have modified it (Page 3, line 107).
108 – “ppm” is not an unit of SI.
Thank you for your constructive comments. In accordance with the Reviewer-1’s comments, we have modified it from “ppm” to “%” (Page 3, line 111 and 131; Page 10, line 233; Page 13, line 380).
Fig. 1A and 1B should be presented as separated single Figures.
We express our deep thanks for your comments. In accordance with the Reviewer-1’s comments, we have separated Fig. 1A to Fig. 1 and Fig 1B to Fig. 2. With this modification, number of other Figures were modified.
Correspondence Figures
Fig. 1A → Fig. 1
Fig. 1B → Fig. 2
Fig. 1C-E → Fig. 3
Fig. 2 → Fig. 4
Fig. 3 → Fig. 5
Fig. 4 → Fig. 6
Fig. 5 → Fig. 7
Fig. 6 → Fig. 8
Fig. 7 → Fig. 9
Information about the used statistical test should be removed from figures description. “4.9 Statistical Analysis” informs about used test.
Thank you for your valuable comments. In accordance with the Reviewer-1’s comments, we have removed the information of statistical analysis in all Figures.
Figure 4 B – Please check significance of the difference for microarray analysis for BANK 1.
Thank you for your valuable comments. We have confirmed the significant difference of protein expression of BANK1 between groups 1 and 2 once again (Figure 6B). However, there was no significant difference on it.
235 – It should be “… activity [20].”
Thank you for your valuable comments. In accordance with the Reviewer-1’s comments, we have added “activity” (Page 10, line 243).
247-261 – This part should belong rather to “Results” instead to “Discussion”.
Thank you for your constructive comments. In accordance with the Reviewer-1’s comments, we have modified this part of Discussion (Page 10, line 255-258). In addition, we have modified Results (Page 5, line 160-161 and line 163-165).
318 – I suggest to add “Conclusions” in this place.
Thank you for your valuable comments. In accordance with the Reviewer-1’s comments, we have made the Conclusions section (Page 12, line 315).
539 – “Cancer Cell” with italic, “2005” with bold.
Thank you for finding our mistake. In accordance with the Reviewer-1’s comments, we have changed it to Italic and Bold (Page 16, line 542).
565 – Journal title abbreviation is needed.
Cancer Genomics Proteomics [Ref.27,28] is abbreviated title. The official name is “Cancer Genomics & Proteomics”.
587, 607 –It should be “PLoS ONE”.
Thank you for finding our mistake. In accordance with the Reviewer-1’s comments, we have modified it (Page 17, line 596).
We strongly appreciate again for the Reviewer-1’s comments, regarding on our paper.
We feel that the revised manuscript is a suitable response to the comments, and is significantly improved over the initial submission. We trust that is now suitable for publication in International Journal of Molecular Sciences.
Thank you in advance for your kind consideration of this paper.
Sincerely yours,
Masaru Terasaki, Ph.D.
Department of Health and Environmental Sciences, School of Pharmaceutical Sciences, Health Sciences University of Hokkaido
1757 Kanazawa, Ishikari-Tobetsu, Hokkaido
061-0293, Japan
Phone: +81-133-23-1211
Fax: +81-133-23-1669
E-mail: terasaki@hoku-iryo-u.ac.jp
Reviewer 2 Report
Manuscript Revision
Title: Fucoxanthin prevents pancreatic tumorigenesis in C57BL/6J mice that received allogenic and orthotopic transplantations of the cancer cells.
Personally, I find the text interesting. The authors have carried out an interesting work on evaluate the effect of diets rich in fucoxanthin on the prevention of pancreatic tumorigenesis in mice. Furthermore, this study presents the basis for promising future studies. However, the manuscript presents a series of drawbacks that need to be corrected before the manuscript can be published. In the following lines I will explain the main mistakes found.
Line 75. They look like two totally different texts from here. The union between the two is missing. It is a shame since the information provided is very interesting and with this small mistake it loses a lot of quality
Line 108. How was this concentration of fucoxanthin obtained from the oil?
Line 110 - 111. Weren't the results compared with diets that did not have fucoxanthin?
Figures are reflected long after they first appear named in the text.
Figure 1. The image quality of the molecule structure is not adequate.
Figure 2. letter size of each image is not equal. Therefore, some attract more attention than others.
Line 156. Supplementary Table? There is no supplementary material, so I cannot rate this information.
Figure 3. Image C cannot be read. Is too small.
Figure 7. the quality of this figure is not good enough.
Line 391. “μL” instead of “μl”. Correct this error throughout the section. The abbreviation for liters must always be capitalized.
There is no section at the end of the article that describes that this test respects the ethical protocols of animal tests.
Author Response
Dec 3, 2021
Dear Ms. Mischa Wang
Assistant Editor, International Journal of Molecular Sciences
Manuscript ID.: ijms-1483444
Ms.Title: Fucoxanthin prevents pancreatic tumorigenesis in C57BL/6J mice that received allogenic and orthotopic transplantations of the cancer cells
Thank you for your email of Nov 30, 2021, regarding our manuscript, “Fucoxanthin prevents pancreatic tumorigenesis in C57BL/6J mice that received allogenic and orthotopic transplantations of the cancer cells”, and the valuable comments of the reviewers. The changed parts in the manuscript file are marked in the red with Track Changes.
Please find enclosed the revised manuscript in Word format (file name: ijms-1483444 with Track Changes).
Comments from the editors and reviewers:
Reviewer 2: Personally, I find the text interesting. The authors have carried out an interesting work on evaluate the effect of diets rich in fucoxanthin on the prevention of pancreatic tumorigenesis in mice. Furthermore, this study presents the basis for promising future studies. However, the manuscript presents a series of drawbacks that need to be corrected before the manuscript can be published. In the following lines I will explain the main mistakes found.
Line 75. They look like two totally different texts from here. The union between the two is missing. It is a shame since the information provided is very interesting and with this small mistake it loses a lot of quality.
We express our deep thanks for your comments. In accordance with the Reviewer-2’s comments, we have modified this sentence (Page 2, line 77).
Line 108. How was this concentration of fucoxanthin obtained from the oil?
Thank you for your valuable comments. This concentration is Fx weight per MF chow.
In accordance with the Reviewer-1 and 2’s comments, we have changed Fx (3,000 ppm) to Fx (0.3%). In addition, we have modified the sentence in Materials and Methods (Page 13, Line 380).
Line 110 - 111. Weren't the results compared with diets that did not have fucoxanthin?
We express our deep thanks for your comments and apologize for the confusion. The diet without Fx is control diet. Mice in group 2 were served the control diet. We have modified the sentence easier to understand (Page 3, line 114).
Figures are reflected long after they first appear named in the text.
Thank you for your constructive comments. We have separated Figures as bellow.
Correspondence Figures
Fig. 1A → Fig. 1
Fig. 1B → Fig. 2
Fig. 1C and 1D → Fig. 3
Fig. 2 → Fig. 4
Fig. 3 → Fig. 5
Fig. 4 → Fig. 6
Fig. 5 → Fig. 7
Fig. 6 → Fig. 8
Fig. 7 → Fig. 9
With this modification, Figure 1 and 2 moved to a nearby location in the text
Figure 1. The image quality of the molecule structure is not adequate.
Thank you for finding our mistake. In accordance with the Reviewer-2’s comments, we have replaced it to higher resolution (Figure 1, Page 2 line 75; Figure 2, Page 3 line 129; Figure 3, Page 4 line 135).
Figure 2. letter size of each image is not equal. Therefore, some attract more attention than others.
Thank you for your constructive comments. In accordance with the Reviewer-2’s comments, we have modified Figure 4 (Page 5, line 144).
Line 156. Supplementary Table? There is no supplementary material, so I cannot rate this information.
In first submission, we sent Supplementary Table S1 and S2 to Editor, for reviewers to see it.
I will ask Editor to send Supplementary Tables to Reviewers again.
Figure 3. Image C cannot be read. Is too small.
Thank you for finding our mistake. In accordance with the Reviewer-2’s comments, we have replaced it to higher resolution (Figure 5C, Page 6 line 170).
Figure 7. the quality of this figure is not good enough.
Thank you for finding our mistake. In accordance with the Reviewer-2’s comments, we have replaced it to higher resolution (Figure 9, Page 11 line 272).
Line 391. “μL” instead of “μl”. Correct this error throughout the section. The abbreviation for liters must always be capitalized.
Thank you for finding our mistake. In accordance with the Reviewer-2’s comments, we have modified it in Material and Method section.
There is no section at the end of the article that describes that this test respects the ethical protocols of animal tests.
Thank you for finding our mistake. In accordance with the Reviewer-2’s comments, we have added the Institutional Review Board Statement (Page 14 line 493-496).
We strongly appreciate again for the Reviewer-2’s comments, regarding on our paper.
We feel that the revised manuscript is a suitable response to the comments, and is significantly improved over the initial submission. We trust that is now suitable for publication in International Journal of Molecular Sciences.
Thank you in advance for your kind consideration of this paper.
Sincerely yours,
Masaru Terasaki, Ph.D.
Department of Health and Environmental Sciences, School of Pharmaceutical Sciences, Health Sciences University of Hokkaido
1757 Kanazawa, Ishikari-Tobetsu, Hokkaido
061-0293, Japan
Phone: +81-133-23-1211
Fax: +81-133-23-1669
E-mail: terasaki@hoku-iryo-u.ac.jp